# Multi-AP and Test Point Accuracy of the Results in WiFi Indoor Localization

**DOI:** 10.3390/s22103709

**Published:** 2022-05-12

**Authors:** Shuyu Li, Sherif Welsen, Vladimir Brusic

**Affiliations:** 1Department of Electrical and Electronic Engineering, University of Nottingham Ningbo China (UNNC), 199 Taikang East Road, Ningbo 315100, China; shuyu.li@nottingham.edu.cn; 2School of Computer Science, University of Nottingham Ningbo China (UNNC), 199 Taikang East Road, Ningbo 315100, China; vladimir.brusic@nottingham.edu.cn

**Keywords:** channel state information, indoor localization accuracy, joint localization, WiFi

## Abstract

WiFi-based indoor positioning has attracted intensive research activities. While localization accuracy is steadily improving due to the application of advanced algorithms, the factors that affect indoor localization accuracy have not been sufficiently understood. Most localization algorithms used in changing indoor spaces are Angle-of-Arrival (AoA) based, and they deploy the conventional MUSIC algorithm. The localization accuracy can be achieved by algorithm improvements or joint localization that deploys multiple Access Points (APs). We performed an experiment that assessed the Test Point (TP) accuracy and distribution of results in a complex environment. The testing space was a 290 m${}^{2}$ three-room environment with three APs with 38 TPs. The joint localization using three APs was performed in the same test space. We developed and implemented a new algorithm for improved accuracy of joint localization. We analyzed the statistical characteristics of the results based on each TP and show that the local space-dependent factors are the key factors for localization accuracy. The most important factors that cause errors are distance, obstacles, corner locations, the location of APs, and the angular orientation of the antenna array. Compared with the well-known SpotFi algorithm, we achieved a mean accuracy (across all TPs) improvement of 46%. The unbiased joint localization median accuracy improved by 20% as compared to the best individual localization.

## 1. Introduction

The rapid development of wireless communication technology and the omnipresence of mobile devices enable the rapid growth of Location-Based Services (LBSs) [1,2]. Outdoor LBSs, such as the Global Navigation Satellite System (GNSS), are used for applications such as navigation, emergency services, timekeeping, or uses for military or geodesy purposes [3,4]. Indoor LBSs complement outdoor LBSs. For example, indoor LBSs can be used to track assets, build management, provide indoor location information for emergency services, and navigate customers in shopping centers [5,6,7]. In healthcare, applications include monitoring patients in nursing homes, tracking Alzheimer’s patients, monitoring the activities and movements of rehabilitating patients, or improving the safety of elderly patients [8,9,10].

The outdoor LBSs cannot be used indoors due to blocked signals. This is because of the additional challenges such as multipath effects, Non-Line-of-Sight (NLoS), moving humans or objects, ambient noise, and electromagnetic interference, which need to be addressed [11]. Reliable Indoor Positioning Systems (IPSs) must satisfy key requirements:

Accuracy: Accuracy is a key improvement target for most indoor LBSs’ research [1]. Improving the indoor LBSs can be addressed by signal processing, additional hardware, model-based analysis, or new communication protocols [12,13,14,15,16,17].

Availability: IPSs should cover all locations within the serviced indoor area at all times. Preferably, IPSs should work with widely available devices, such as mobile phones or WiFi [12]. Some solutions, such as Ultra-Wide Band (UWB) [18] or Visible Light Communication (VLC) [15,19] systems, require additional hardware, limiting their usability.

Scalability: IPSs should perform well when the indoor environment changes. Most Machine Learning (ML) or fingerprinting-based IPSs lack reliability in changing environments [20] because these methods are highly dependent on pre-training defined by particular environment settings.

Cost: IPSs’ implementation should not require a high capital cost. The system should not include any additional infrastructure or expensive equipment on the server and user sides. The system should be easily maintained without investing much labor and hardware maintenance.

Other requirements: Requirements specific to the use scenarios and applications may include update rates, privacy, security, and user interfaces, among others [21].

The availability, ubiquity, scalability, and low cost of Wireless Local Area Network (WLAN) systems make WiFi-based indoor localization a competitive solution for IPSs. The protocols defined in the IEEE802.11a/n standards make the Channel State Information (CSI) of WiFi signals readily available [22]. CSI provides stable and feature-rich channel information, which enables the estimation of the AoA. The AoA and the signal strength information enable the estimation of the source position. The AoA is calculated by applying algorithms, such as MUSIC [23]. The AoA-based localization algorithms, such as SpotFi [13], can achieve a decimeter-level median localization accuracy in an office environment.

Factors affecting the accuracy of indoor localization of specific location points are poorly understood. Most localization performance assessments utilize median accuracy [13,14,24,25,26], but they do not provide detailed spatial assessment maps. We are particularly interested in possible blind localization points in indoor spaces and the reasons for their existence. Therefore, we performed extensive data analysis and factor analysis based on the experiment. The sources of errors may include doors, obstacles, human activities, space settings, electromagnetic interferences, and the number of AP, among others [27]. To better understand the factors that limit localization accuracy and improve performance, we focused on better utilizing available information, optimizing localization algorithms, and performing a comprehensive localization testing experiment.

Utilization of information: Most of the information available for signal processing is not considered or fully utilized. For example, AoA-based approaches such as ArrayTrack [25] and SpotFi [13] do not consider the amplitude of the AoA spectrum for the LoS identification. The amplitude of the AoA spectrum can reflect the power of different paths for better LoS identification. In addition, the variance of RSS is also neglected in most reported algorithms. We used additional information to improve IPS performance.

Optimization by joint localization: Most of the known methods are not optimized for multi-AP localization. For instance, ROArray [14] locates the target among multiple APs by minimizing an RSS-based objective function. SpotFi [13] and MaTrack [28] use similar approaches to find the optimal solution from a large number of results. These methods are not computationally efficient in performing joint localization.

Adequacy of TP-based analysis: Most of the reported methods in WiFi indoor localization do not provide a detailed analysis of multiple TPs and scenarios. Some reports use simulations to assess the accuracy of localizations [29] or actual point localization error [30], but the TP-based error analysis is largely lacking in available reports. The TP-based error maps enable an analysis of factors that affect localization accuracy. We performed a localization experiment and accuracy analysis at each of the 38 TPs within a test space (a multi-room space with three APs).

We intended to compare the performance of different localization algorithms. However, some available algorithms [14] are proprietary and do not provide the source code. Other studies [31] involved custom-designed hardware, which could not be reproduced for our study. Therefore, it was not possible to compare the performance of the majority of the reported algorithms. The well-known SpotFi algorithm was chosen as the benchmark; it is available as open-source [32] and uses a commodity WiFi device.

We previously developed MoLA, a Multi-step Optimization Localization Algorithm, and compared the performance of MoLA and SpotFi for a single AP. We showed that the multi-step optimization used in MoLA improved the median accuracy of indoor localization as compared with SpotFi (0.96 m vs. 1.93 m) in the test space [33]. To enable further comparisons and encourage knowledge sharing, we provided MoLA to the community as open-source [34]. We extended the initial version of MoLA and used it to perform a multi-AP assessment of indoor localization accuracy by using multiple APs. For joint estimation, MoLA locates the target by weighting different APs based on the variance information of the RSS. MoLA adaptively combines multiple APs to improve the localization results. The flowchart of the proposed method is shown in Figure 1. There are three main contributions of this work:Developed and implemented a new self-adaptive multi-AP localization algorithm that improves MoLA system performance;Performed a comprehensive statistical analysis of individual TP localization accuracy;Analyzed the effects of TP location in the room setup on localization accuracy. We identified and described key factors that affect the individual TP and the overall localization accuracy.

Section 2 describes related works in CSI-based indoor localization. The technical background of this study is presented in Section 3. Section 4 describes the design of MoLA and its multi-AP localization extension. Section 5 describes the hardware platform and the experimental results in different cases. Section 6 analyzes the factors of localization errors, and Section 7 concludes this paper.

## 2. Related Work

### 2.1. AoA-Based Systems

With the development of MIMO-OFDM technology, AoA approaches based on CSI became popular solutions in the indoor localization field. All AoA-based algorithms deploy the conventional MUSIC algorithm [23] to estimate the AoA. The AoA-based IPS achieves decimeter accuracy. The ArrayTrack system [25] uses the phase shift of the received signal with 6–8 antennas. A system called Phaser [26] synchronizes two network cards by sharing one antenna. Phaser achieves decimeter accuracy with five antennas. The Ubicarse system [24] is a new Synthetic Aperture Radar (SAR) formulation for locating the target within an indoor space. Ubicarse requires at least two antennas fixed at the localization target and the rotation of antennas around the vertical axis while positioning. However, hardware modifications are required for operating ArrayTrack, Phaser, or Ubicarse. The SpotFi system [13] introduces an indoor localization system that does not need any hardware modifications to improve usability. It includes a “super-resolution” AoA algorithm to estimate the AoA by creating a smooth CSI matrix. SpotFi utilizes a clustering method and a likelihood algorithm to identify the LoS path to deal with the multipath effect. Unlike other MUSIC-based AoA estimation methods, ROArray [14] proposes a robust indoor localization system using a sparse recovery method. It can obtain a median localization accuracy in a meter range in low Signal-to-Noise Ratio (SNR) scenarios. Multi-AP localization systems such as SpotFi and ROArray have high computational costs.

### 2.2. Range-Based Systems

FILA, a range-based system [16], uses a fast supervised training algorithm to refine the model using the amplitude information of the CSI. FILA cannot be deployed directly in the spaces where the settings change. The Chronos system [12] uses a Time-of-Flight (ToF) approach with a single AP. It uses the characteristics of the bandwidth and time to imitate the wideband system. Chronos achieves a moderate median accuracy using a single AP. However, this method requires a specialized protocol for frequency sweeping and software modifications on commercial WiFi devices. A single AP system S-Phaser deploys a phase calibration solution called the Interpolation Elimination Method (IEM) [35]. It determines the target location by the Broadband Angle Ranging (BAR) algorithm. S-Phaser can obtain a low median accuracy of 1.5 m in an LoS environment. MaTrack [28] and LiFS [36] are device-free passive indoor localization methods. They locate targets within a specific range by observing the AoA and ToF of one or multiple pairs of signals. However, these methods have limited abilities to locate multiple objects or targets with no LoS and are very sensitive to environmental changes.

### 2.3. Learning-Based Systems

FIFS [37] proposed an indoor fingerprint localization system based on the CSI. It builds the fingerprint database by measuring the signal amplitude on multiple antennas. However, the amplitude data are not informative, and the phase information in the CSI is not fully utilized. To overcome this problem, PhaseFi [38] proposed a localization algorithm using calibrated phase data features. It designs a deep network to train the data and optimize the computational cost by using a greedy learning algorithm. The localization error is about 1 m in the open environment and 2 m in a complex environment. DeepFi [39] uses deep learning methods to deal with WiFi signals. However, this approach requires constructing a fingerprint database and the collection of information on each location. This method requires high labor cost and effort and is vulnerable to changes in the spatial configuration and obstacles.

### 2.4. Single-AP MoLA

MoLA collects CSI from commercial devices and uses a phase calibration algorithm to eliminate phase errors. MoLA then applies the spatial smoothing algorithm [40] and the I-MUSIC algorithm [41] to find possible AoAs. The Sequential Quadratic Programming (SQP) algorithm [42] is used to solve this optimization problem. In this process, the MDL equation [43] is used to estimate the number of LoS signals from different directions. To further deal with multipath effects and noise, MoLA first groups the AoAs from multiple packets and then builds a new estimator to identify the LoS cluster.

## 3. Background

Assume that the receiver has a Uniform Linear Array (ULA), consisting of *M* antennas spaced at a distance *d*. As the signal emitted from the transmitter reaches the antenna array, an AoA with an angle θ is generated due to the distance difference existing between the two adjacent antennas (Figure 2). In indoor environments, obstacles often reflect signals, resulting in multipath effects. Assuming that there are *L* propagation paths, the phase shift of the *l*th path at each antenna is [44]:(1)$$\Lambda \left(\theta_{l}\right)=e^{-j2\pi \times d\;\cos\left(\theta_{l}\right)\times \frac{f}{c}},$$
where *f* is the signal frequency and *c* is the speed of light. In an OFDM system, each subcarrier is modulated into a different frequency, which causes a delay. The phase shift caused by the delay function is:(2)$$\Gamma \left(\tau_{l}\right)=e^{-j2\pi \times f_{\delta }\times \tau_{l}},$$
where $f_{\delta }$ is the frequency interval between subcarriers and $\tau_{l}$ is the transmission delay of the *l*th path. Combining the phase shift $\Lambda (\theta_{l})$ and phase shift $\Gamma (\tau_{l})$, the steering vector matrix with *K* subcarriers’ transmission delays can be written as:(3)$$\begin{array}{ccc} a(\theta_{l},\tau_{l})= & [ & \underset{\mathrm{Antenna}}{\underbrace{1,\ldots ,\Gamma {\left(\tau_{l}\right)}^{K-1}}},\\ \; & \; & \underset{\mathrm{Antenna}}{\underbrace{\Lambda \left(\theta_{l}\right),\ldots ,\Lambda \left(\theta_{l}\right)\Gamma {\left(\tau_{l}\right)}^{K-1},}}\ldots ,\\ \; & \; & \underset{\mathrm{Antenna}\;M}{\underbrace{\Lambda {\left(\theta_{l}\right)}^{M-1},\ldots ,\Lambda {\left(\theta_{l}\right)}^{M-1}\Gamma {\left(\tau_{l}\right)}^{K-1}}}{]}^{T}. \end{array}$$

After obtaining the steering vector, we calculate the auto-correlation matrix of the data matrix and apply the MUSIC algorithm. The data matrix is obtained from the device, as a CSI matrix H, expressed as:(4)$$H=\left[\begin{array}{cccc} {csi}_{1,1} & {csi}_{1,2} & \cdots & {csi}_{1,K}\\ {csi}_{2,1} & {csi}_{2,2} & \cdots & {csi}_{2,K}\\ \vdots & \vdots & \ddots & \vdots \\ {csi}_{M,1} & {csi}_{M,2} & \cdots & {csi}_{M,K} \end{array}\right].$$

The auto-correlation matrix $R_{x}$ is defined by [44]:(5)$$R_{x}=E\left[HH^{H}\right].$$

After performing the Eigenvalue Decomposition (EVD) of $R_{x}$, the noise subspace $E_{N}$ can be obtained. The MUSIC algorithm is used to find the AoA and ToF:(6)$$P{\left(\theta_{l},\tau_{l}\right)}_{MUSIC}=\frac{1}{a\left(\theta_{l},\tau_{l}\right)E_{N}E_{N}^{H}a\left(\theta_{l},\tau_{l}\right)}.$$

## 4. MoLA: Design and Optimization

To localize the target, MoLA performs phase correction, AoA estimation, and identification of the LoS path. Here, we report a new adaptive joint localization algorithm. A detailed description of MoLA is available in [33]. This section provides a brief description of MoLA.

### 4.1. Pre-Processing of CSI

Due to imperfections in the hardware, the received CSI signal array produces phase shifts. The errors can be caused by Packet Detection Delay (PDD), Sampling Frequency Offset (SFO), Center Frequency Offset (CFO), and some other causes [45]. All these factors will impact the AoA estimation, so we modeled the phases as described in [46]. The measured phase $\varphi_{m,k}^{\left(i\right)}$ from the *i*th packet of the *m*th antenna at the *k*th subcarrier can be represented as:(7)$$\varphi_{m,k}^{\left(i\right)}=\phi_{m,k}^{\left(i\right)}-2\pi \times f_{s}\times k\times \delta +\beta +Z,$$
where $\phi_{m,k}^{\left(i\right)}$ is the true phase, $f_{s}$ is the frequency interval of the subcarrier, δ is the time delay at the receiver, β is the non-linear phase offset, and *Z* is the noise. The comparison between the raw and calibrated phases is shown in Figure 3. The calibrated phases identify the narrow range of the true CSI phase (congregation of red points in Figure 3), which otherwise may not be directly identifiable from the raw CSI phase data. The CSI phase calibration is performed as described in Algorithm 1.
**Algorithm 1:** Phase calibration.
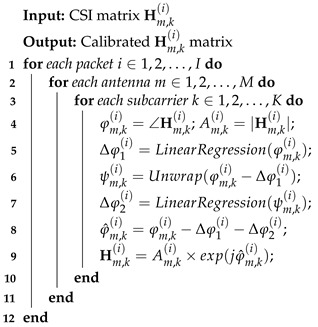


### 4.2. Estimating the AoAs with Improved MUSIC and MDL

#### 4.2.1. The I-MUSIC Algorithm

MoLA estimates the AoA of the incoming wave signal. Due to the complex indoor environment, the signal is coherent and creates multipath effects. To decorrelate the signals as much as possible, MoLA deploys the I-MUSIC algorithm for the smoothed correlation matrix of the signal [40]. The modified correlation matrix $R_{xx}$ is defined by:(8)$$R_{xx}=R_{x}+JR_{x}^{*}J,$$
where
(9)$$J=\left[\begin{array}{cccc} 0 & 0 & \cdots & 1\\ 0 & 0 & 1 & 0\\ \vdots & \vdots & \ddots & \vdots \\ 1 & 0 & \cdots & 0 \end{array}\right].$$
where J is a *K*-order opposed unit matrix and “*” denotes the complex conjugate of the correlation matrix $R_{x}$, as defined in Equation (Equation 5).

#### 4.2.2. Estimating the Number of Signals with MDL

The number of incoherent signals is important in spatial–spectral estimation because it determines the threshold for subspace partition. In SpotFi [13], they used a fixed threshold to determine the subspace. This is an imperfect solution because the indoor environment is often variable. The MDL equation [43] determines the number of incoherent signals required for spatial–spectral estimation. Given that the correlation matrix $R_{xx}$ in Equation (Equation 8) is of order *K* and the number of incoherent signals is *p*, then *p* is determined by minimizing the estimator:(10)$${\hat{p}}_{MDL}=\underset{p\in \{0,\cdots ,K-1\}}{\arg\min}MDL(p),$$
where
(11)$$\begin{array}{cc} MDL(p) & =\left(K+M-1\right)\times \left(K-p\right)\log\left(\frac{\frac{1}{K-p}\times \sum_{k=1}^{K-p}\lambda }{{\left(\prod_{k=1}^{K-p}\lambda \right)}^{\frac{1}{K-p}}}\right)\\ \; & +\frac{p}{2}\times \left(2K-p+1\right)\times \log\left(K+M-1\right), \end{array}$$
where p∈{0,⋯,K−1} and the integer *k* that makes MDL(p) smallest is the number of estimated incoherent signals. λ is the diagonal vector of the eigenvalue matrix of the $R_{xx}$ after EVD, which is ordered from largest to smallest:(12)$$\lambda ={\left[\lambda_{1},\lambda_{2},\cdots ,\lambda_{k},\cdots ,\lambda_{K}\right]}^{T},\;\lambda_{1}\geq \lambda_{2}\geq \ldots \geq \lambda_{k}\geq \ldots \geq \lambda_{K}.$$

### 4.3. Identification of the LoS

After performing AoA estimation on multiple packets, we use a clustering algorithm to cluster the possible AoAs. To identify the LoS accurately, we developed and implemented a new LoS estimator. It calculates the number of AoAs in each cluster, the compactness of the clusters, and the transmission delay of each cluster. The estimator can be expressed as:(13)$${WE}_{n}=w_{n}\times \exp\left(C_{N}N_{n}-C_{P}{\bar{CP}}_{n}-C_{\tau }{\bar{\tau }}_{n}\right),$$
where $N_{n}$ is the number of points in the cluster *n*. ${\bar{CP}}_{n}$ is the compactness, and ${\bar{\tau }}_{n}$ is the mean delay of transmission of the cluster *n*. $C_{N}$, $C_{P}$, and $C_{\tau }$ are the scale parameters used to bring the exp factor into the (0,10) range. $w_{n}$ is the mean weight of cluster *n*.

### 4.4. Target Localization

The distance between the target and the receiver can be obtained by minimizing the following equation *D*:(14)$$D=\arg\min_{R}\sum_{i=1}^{I}\left[{\left({\hat{PR}}_{i}-{PR}_{i}\right)}^{2}\right],$$
where ${\hat{PR}}_{n}$ and ${PR}_{n}$ are the observed and estimated RSS values. The coordinate of the target $\left(\hat{x},\hat{y}\right)$ can be obtained by performing geometric calculations with the previously obtained direct path $\bar{\theta }$ and distance *D*. The process described in Sections (B), (C), and (D) is formalized in Algorithm 2.
**Algorithm 2:** AoA Estimation and localization.
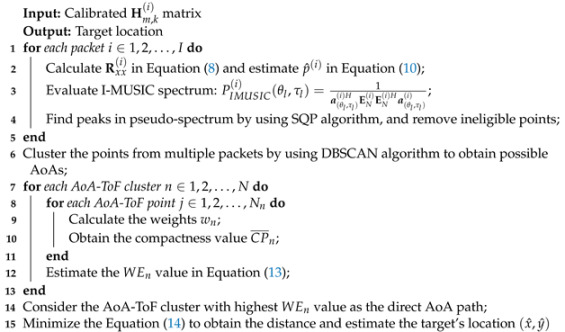


### 4.5. Multiple AP Localization

#### 4.5.1. Conventional Trilateration with the Kalman Filter

In the existing RSS-based trilateration localization methods, the RSS signal is often processed using a Kalman Filter (KF) [47]. The wireless signal transmission model is:(15)$$d=d_{0}10^{\frac{P_{r,dB}\left(d_{0}\right)-P_{r,dB}\left(d\right)}{10\eta }},$$
where $P_{r,dB}\left(d_{0}\right)$ and $P_{r,dB}\left(d\right)$ are the received signal power at the TP, which have distances $d_{0}$ and *d*. η is the path loss exponent of the signal attenuation model.

This model will be used as a benchmark for the analysis of the results of multi-AP localization.

#### 4.5.2. Characteristics of the RSSI among Different Locations

The previous work was extended to joint localization by multiple APs. First, we need to investigate the characteristics of the RSSI at different locations. The OFDM provides rich information on the RSSI measurements; each subcarrier has a unique RSSI value because of different multipath fading channels. The RSSI value for each subcarrier varies at the same TP, but varies with different locations (Figure 4a). Therefore, the RSSI variance of each packet at different locations reflects the changes in the environment better than a single RSSI value (Table 1).

To show the high correlation between environment changes and RSSI variance, we collected the RSSI value through 38 different locations from AP${}_{1}$. Each bar represents the variance of the RSSI for each subcarrier throughout multiple packets (Figure 4b). RSSI variance is relatively high among Points 28–31, which are all NLoS locations when using AP${}_{1}$.

#### 4.5.3. Weight Calculation and Multi-AP Localization

We propose a new multi-AP localization algorithm based on the observed relationship between the RSSI variance and specific locations (Figure 4b). This algorithm calculates the weight of each AP from the variance of RSSI measurements at each target location and then assigns these weights for multi-AP localization (Equation (Equation 16)). Locations with low RSSI variance contribute significantly to the results, while locations with large RSSI variance (rich NLoS environments) contribute very little to the final results. A better signal strength usually means that the target is more likely to be in an LoS or less multipath environment for observed AP. TPs with strong RSSI weights will assign higher weights to the receiving AP and vice versa.

This approach utilizes the multidimensional information of the RSSI and adaptively estimates the target location through multiple APs. The joint estimated location $\left(\hat{X},\hat{Y}\right)$ is formulated by:(16)$$\left(\hat{X},\hat{Y}\right)=\left(\sum_{q=1}^{Q}w_{q}{\hat{x}}_{q},\sum_{q=1}^{Q}w_{q}{\hat{y}}_{q}\right),$$
where ${\hat{x}}_{q}$ and ${\hat{y}}_{q}$ are the estimated coordinates from the *q*th AP, *Q* is the number of APs, and $w_{q}$ is the weight. It can be calculated by:(17)$$\begin{array}{cc} w_{q} & =\frac{\frac{1}{M(V_{q})}}{\frac{1}{M(V_{1})}+\frac{1}{M(V_{2})}+\cdots +\frac{1}{M(V_{q})}}\\ \; & =\frac{1}{M(V_{q})}\times \frac{\prod_{q=1}^{Q}M(V_{q})}{\sum_{q=1}^{Q}M(V_{q})}, \end{array}$$
where M(·) represents the median value and $V_{q}$ is the variance column vector from *K* subcarriers of the *q*th AP:(18)$$M\left(V_{q}\right)={\left[var_{1}^{\left(q\right)},var_{2}^{\left(q\right)},\cdots ,var_{k}^{\left(q\right)}\right]}^{T},$$
where $var_{k}^{\left(l\right)}$ is the RSSI variance from *I* packets:(19)$$M\left(V_{q}\right)=Var\left(RSSI_{k}^{\left(1\right)},RSSI_{k}^{\left(2\right)},\cdots ,RSSI_{k}^{\left(i\right)}\right).$$

The process described in Section E is formalized in Algorithm 3.
**Algorithm 3:** Adaptive multi-AP localization.
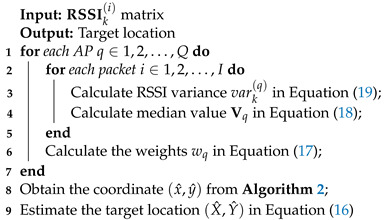


## 5. Experimental Evaluation

### 5.1. Implementation

We deployed the experimental platform (Figure 5a) in a large 290 m${}^{2}$ conference room to validate MoLA, with SpotFi serving as a benchmark. For the hardware part, we used an Industrial Control Computer (ICC) as the AP, which was equipped with an Intel 5300 Network Interface Card (NIC) (Figure 5b). Each ICC was equipped with three antennas, spaced 2.8 cm, and placed in three different locations in the conference room. The antennas were mounted on the ICC aluminum housing (Figure 5c). All NICs were set to monitor mode (bandwidth of 40 MHz, operating in the 5 GHz band). We broadcast each TP and collected data on the three ICCs.

We performed six measurements for each TP. Three measurements were performed on one day at a fixed time (9:00–9:45, 12:00–12:45, and 19:00–19:45). The order of measurements was from TP1 to TP38 (Figure 5a). Measurements were repeated the following day at the same time. At each TP, we sent a total of 300 packets with a transmission interval of 100 ms. Then, we compared the results with the coordinates obtained using a laser range finder. We calculated the angle between each TP and AP’s antenna within the LoS path as the AoA’s ground truth since the position of the APs and TPs was fixed in the room. The impact of the input packet number (between 5 and 300) on the localization error was analyzed, as described in Section 6.

### 5.2. TP-Based Localization Accuracy

We conducted experiments at different locations and in different scenarios to evaluate the accuracy of MoLA. TPs 1–27 in the large room (Figure 5a) represent the LoS scenario, while the other TPs represent the NLoS scenario. The overall localization performance of the two systems was compared, and the results are shown in Figure 6a. MoLA achieved a median error of 0.9 m, while SpotFi had a median error of 1.9 m under the AP${}_{1}$ scenario.

We then analyzed the different blocking cases with the LoS and NLoS scenarios. Figure 6b depicts the localization error without any obstructions, where MoLA achieved a median accuracy of 0.7 m and SpotFi achieved 1.5 m. When the TP was moved in an enclosed environment, the median error increased to 1.7 m for MoLA and 3.3 m for SpotFi (Figure 6b).

The accuracy varied between TPs. We calculated the standard deviation (σ) of the localization results for each TP. The largest standard deviation of the localization error for AP${}_{1}$ was TP20 (3.9 m ± 1.1 m). For AP${}_{2}$, the largest was TP33 (5.9 m ± 1.6 m). For AP${}_{3}$, the largest was TP6 (2.1 ± 2.2 m). This illustrates that the overall localization error is related to the position of the AP, while the individual localization error is related to the position of the TP.

### 5.3. AoA Comparison

To verify the algorithm’s reliability, we compared the AoA estimation error for each TP with the ground truth AoA of each TP. The comparison showed that MoLA had higher accuracy (Figure 6c). In the LoS case, the median AoA error was 6 degrees for MoLA and 10 degrees for SpotFi. The AoA error in the NLoS case was 13 degrees for MoLA and 24 degrees for SpotFi. These results indicate that the clustering algorithm and the weighted estimator Equation (Equation 13) effectively discriminated the LoS path from multiple incident signals. Furthermore, these improvements may be due to the correct partitioning of the subspace by the MDL algorithm. SpotFi uses the conventional MUSIC algorithm with a fixed threshold to estimate the spectrum. However, MUSIC is sensitive to the noise subspace $E_{N}$, and a fixed threshold does not provide optimized results.

### 5.4. Multi-AP Localization Accuracy

For other APs, MoLA achieved a median error of 2.3 m at AP${}_{2}$, compared to 3.4 m for SpotFi. In contrast, at AP${}_{3}$, in the NLoS scenario, MoLA achieved a median error of 1.4 m, while SpotFi degraded to 3.2 m (Figure 7a). Thus, MoLA improved positioning accuracy more than SpotFi, with an average error reduction of 46.2% (50.3% at AP${}_{1}$, 31.3% at AP${}_{2}$, and 57.0% at AP${}_{3}$).

Next, we investigated the error of joint localization. The signal from each TP was broadcast to three APs, and both the CSI and RSSI information was collected. The results showed that MoLA further improved the localization performance through multiple APs (Figure 7b). Compared to the single-AP median errors of 0.9 m, 2.3 m, and 1.4 m, MoLA achieved a median accuracy of 0.7 m, better than any single-AP scenario.

To assess the performance of multi-AP MoLA localization, we compared it with the RSSI trilateration with the KF. However, the path loss exponent (Equation (Equation 15)) is sensitive to the environment. We set the path loss exponent according to the environment and then chose the best one to compare with MoLA. The received signal strength at 1 m is based on the real measured numbers ($P_{r}^{\left(1\right)}\left(d_{0}\right)=-39\;\mathrm{dB},\;P_{r}^{\left(2\right)}\left(d_{0}\right)=-41\;\mathrm{dB},\;P_{r}^{\left(3\right)}\left(d_{0}\right)=-44\;\mathrm{dB}$). The CDF plot (Figure 8a) shows the impact of different path loss exponents on the median localization error. The results indicate that the KF-based trilateration method had the best localization accuracy (2.7 m) when η=2.5 (Figure 8b).

In addition, we further minimized the error by jointly varying the path loss exponent and the received power at 1 m ($P_{r}^{\left(1\right)}\left(d_{0}\right)=-36\;\mathrm{dB},\;P_{r}^{\left(2\right)}\left(d_{0}\right)=-41\;\mathrm{dB},\;P_{r}^{\left(3\right)}\left(d_{0}\right)=-39\;\mathrm{dB},\;\eta^{\left(1\right)}=2.3,\;\eta^{\left(2\right)}=1.7,\;\eta^{\left(3\right)}=2.2$), which finally gave the best median localization error of about 2 m. The CDF plot (Figure 8c) shows that the conventional trilateration method had a median localization error similar to that of the single-AP MoLA. With the number of APs increasing from one to three, localization accuracy improved. We also observed that 80% of the individual TPs had the localization error within 2.5 m with three APs, even within the complex space of 290 m${}^{2}$ with rich multipath. The median value of optimized trilateration with the KF was approximately 2 m, demonstrating the localization accuracy improvement of multi-AP MoLA.

## 6. Analysis of Influencing Factors

### 6.1. Impact of the Packet Number

We investigated the distribution properties of the MoLA results. We found the variance ($\sigma^{2}$) for six different rounds of measurements for each TP, which gave us a total of 38 values. Since six measurements were performed on each TP, we calculated the pooled variance ($S^{2}$) of all 38 points [48]. Figure 9a shows the pooled variance of localization error for various packet numbers. For AP${}_{1}$ and AP${}_{3}$, placed at corner positions, $S^{2}$ decreased as the number of input packets increased. For AP${}_{2}$, placed in the middle of the long wall (Figure 5a), both $S^{2}$ and localization errors were relatively stable; $S^{2}$ was between 0.2 and 0.8, and localization error was approximately 2.5 m for all packets. Overall, the results indicated that the minimum number of packets used for localization should be 50 or 100 (Figure 9b).

The stability of the localization error depends on the AP location (Figure 9a). Higher stability is not necessarily associated with higher localization accuracy. For example, AP${}_{2}$ showed high stability, but low localization accuracy. The analysis of the causes of localization errors offers an insight that can guide the multi-AP system by placing APs in favorable positions, such as corners and places with a rich LoS.

### 6.2. TP-Based Analysis

To better understand the causes of localization errors, we analyzed the results at each TP. To compare the localization error of MoLA and SpotFi, we defined several thresholds, where TR${}_{1}$ is the localization performance of MoLA and TR${}_{2}$ is the localization performance of SpotFi. We identified the TPs with errors less than TR${}_{1}$ as High-Accuracy Localization Points (HA${}_{\mathrm{LP}}$s); TPs with errors between TR${}_{1}$ and TR${}_{2}$ as Median-Accuracy Localization Points (MA${}_{\mathrm{LP}}$s); and TPs with errors greater than TR${}_{2}$ as “Blind Points (BP)” (Table 2). We analyzed localization errors at each TP under three different AP locations (Table 3).

The TP-based error distribution indicated the reasons for higher MA${}_{\mathrm{LP}}$ and BP, generated by several reasons:

Corner Location: For all APs (AP${}_{1}$, AP${}_{2}$, and AP${}_{3}$), the TPs in the corners had high localization errors (Figure 10). The examples include TP19 and TP20 in the large room, TP32 and TP35 in the small room, etc.

Obstacles: For the NLoS cases, TPs without direct paths in the small rooms, the localization accuracy was lower. For the LoS case, a reinforced concrete pillar in the main room was the main obstacle, and the TPs around it or in the area behind its shadowing were affected by the multipath.

Distance: For TPs far away from the AP, the localization accuracy decreased, such as TP 36–38 with AP${}_{1}$. Although the NLoS and antenna orientation also influenced the accuracy, distance was still a key factor that impacted the results. The heat map of three APs and the weighted method were constructed to show the distribution of localization error among different scenarios (Figure 11).

### 6.3. Impact of Antenna Array Orientation

In front of the antenna arrays, the TP within the V-shaped sectors (angular shape with a 120° field of view) showed better localization accuracy than the TPs outside the sector (Figure 10). Examples include TP10 and TP19 in Figure 10a and TP 1–7 in Figure 10b. However, in Figure 10c, TP 28–31 within the V-shaped sector had good localization accuracy even if there was no LoS path. Overall, the results showed that the antenna array orientation defines a narrow V-shaped sector with a high localization accuracy of TPs within the sector. The results showed that covering more TPs within the V-shaped sector of the antenna is expected to yield good localization performance. To obtain higher localization performance, we need to adjust the antennas’ orientation and deploy APs at positions covering long LoS areas to increase localization accuracy in the AoA approach.

### 6.4. Impact of the AP’s Location

The location of the AP can have a significant impact on the localization results (Figure 11). Considering all the results, the localization performance will be strongly influenced by the positioning of the AP in the target localization space. This position should satisfy as much coverage of the LoS area as possible, optimized antenna orientation, and the shortest possible transmission distance. Since the primary role of APs is to ensure normal network communication, optimizing such networks for highly accurate indoor localization becomes a design optimization problem.

### 6.5. Other Considerations

Antenna Configuration: The configuration of the antennas will have some effect on the results, such as the number of antennas, the array aperture, etc. In general, more antennas will provide for higher localization accuracy. However, this will also increase the computational complexity and the cost of the system. The use of the limited number of antennas is still predominant in existing commercial WiFi solutions that provide a large area of network coverage. Increasing the linear array aperture will enhance the resolution of the algorithm to obtain higher SNR spectral peaks. However, the aperture must not be larger than the half-wavelength of the transmitted signal to prevent the occurrence of spurious spectral peaks. In our experiments, we used the half-wavelength criterion for antenna spacing to obtain the best performance.

Co-Channel Interference (CCI): Adjacent and co-channel interference are some of the biggest problems noted in the IEEE 802.11 standard, while channel overlapping is the main cause of CCI [49]. In this work, we used the 5 GHz band to minimize the occurrence of CCI effects. Since our test site was covered by a high density of WiFi devices, which included the 5 GHz band, CCI inevitably generated packet loss and interferences. We collected the results multiple times for each TP to minimize the effects of CCI.

## 7. Conclusions

We extended the original single-AP localization algorithm [33], implemented it for a multi-AP solution, and proposed a novel adaptive localization algorithm with a low computational cost. The median localization accuracy was improved by ∼20% by an unbiased combination of three APs as compared to the best individual AP localization. We conducted an exhaustive TP-based analysis and analyzed the distribution of the results. We identified factors that affect the accuracy of WiFi indoor positioning. The most influential factors were distance, corner locations, obstacles, AP location, and the orientation of the antenna arrays. Additional factors that affect localization accuracy are antenna configurations and co-channel interferences. The heat map of TP locations showed the distribution of localization errors, and it enables factor analysis. The heat map and the information about the distribution of localization errors enable the calibration of localization spaces and promise further improvements in localization accuracy by WiFi devices. The location and orientation of APs for improved localization accuracy can be optimized for further improvement of localization accuracy.

## Figures and Tables

**Figure 1 sensors-22-03709-f001:**
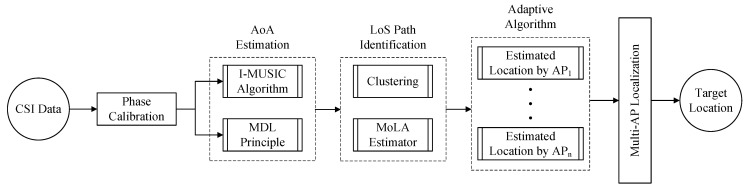
Flowchart of the proposed method.

**Figure 2 sensors-22-03709-f002:**
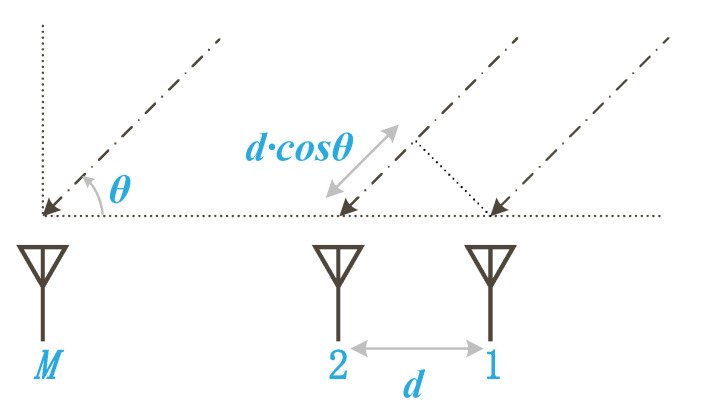
The uniform linear array consists of *M* antennas spaced at a distance *d*. Due to the additional distance dcos(θ) traveled in propagation, the signal is phase-shifted.

**Figure 3 sensors-22-03709-f003:**
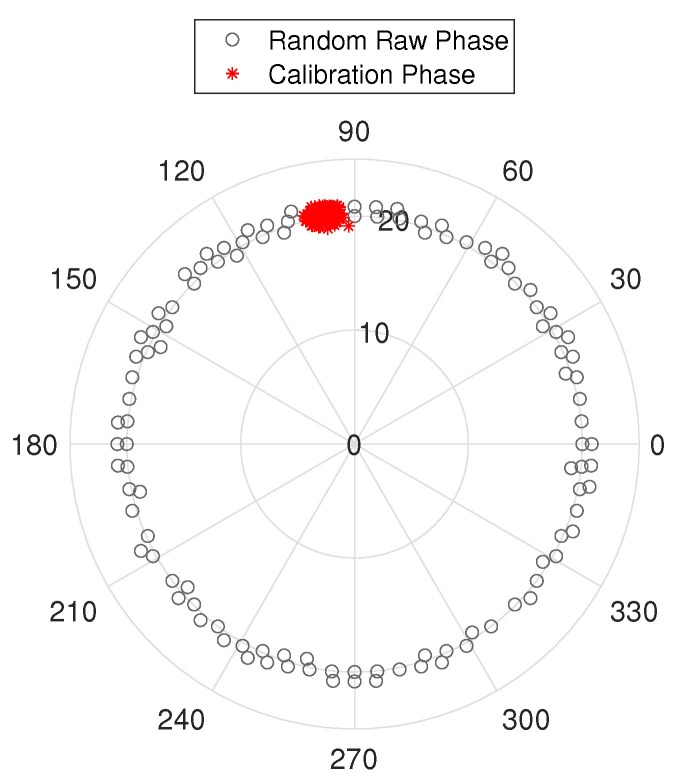
Phase offsets are removed by using the calibration algorithm. In polar coordinates, the phase of the raw signal exhibits a random distribution. After calibration, the real phase shift is shown.

**Figure 4 sensors-22-03709-f004:**
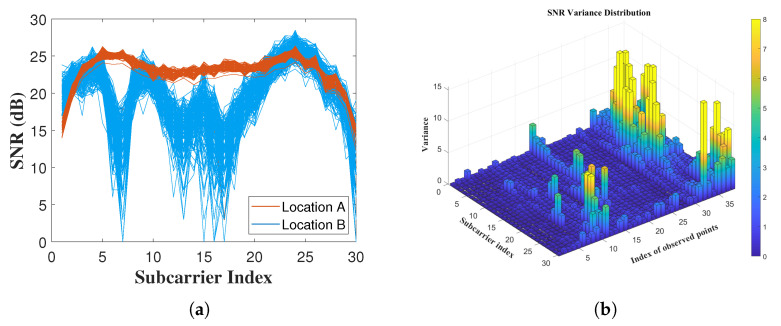
RSSI measurements and variance. (**a**) RSSI of two different locations; (**b**) RSSI variance among all the TPs.

**Figure 5 sensors-22-03709-f005:**
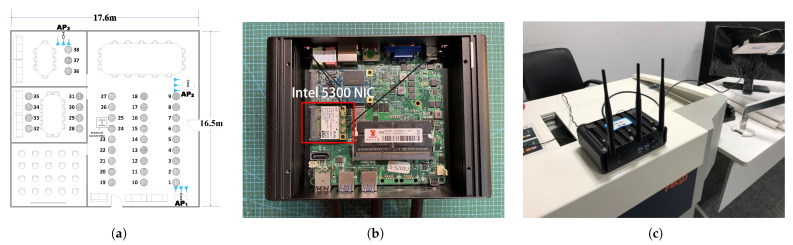
Experimental configurations. (**a**) There are 38 TPs and the blue symbols show the orientation of the antenna array; (**b**) ICC with Intel 5300 NIC; (**c**) practical deployment of APs.

**Figure 6 sensors-22-03709-f006:**
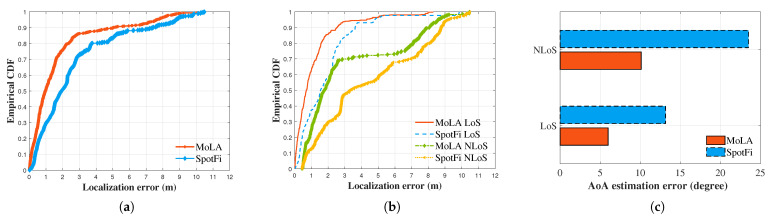
The cumulative distribution function of the localization error between MoLA and SpotFi under the LoS and NLoS scenarios. (**a**) Overall localization error; (**b**) the LoS/NLoS case; (**c**) comparison of the AoA estimation error in degree.

**Figure 7 sensors-22-03709-f007:**
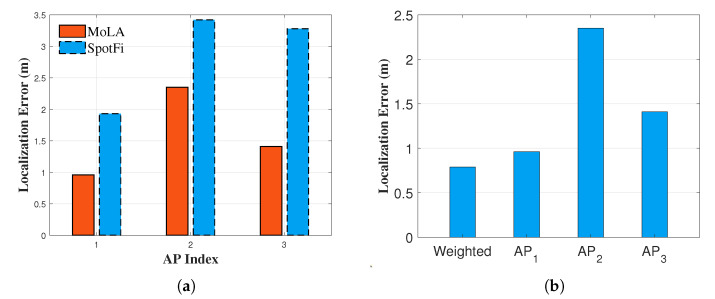
Comparison of localization errors with different APs. (**a**) Localization error of MoLA and SpotFi at three different AP locations; (**b**) localization error using a single AP and the weighted algorithm.

**Figure 8 sensors-22-03709-f008:**
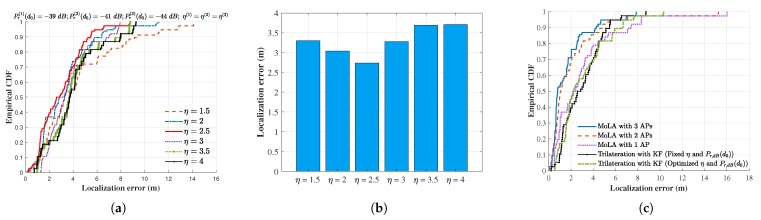
Comparison of localization errors with multi-AP MoLA and optimized trilateration with the KF. (**a**) Impact of different path loss exponents on the localization error of conventional trilateration with the KF; (**b**) localization error with different path loss exponents; (**c**) localization error with multi-AP MoLA and trilateration with the KF.

**Figure 9 sensors-22-03709-f009:**
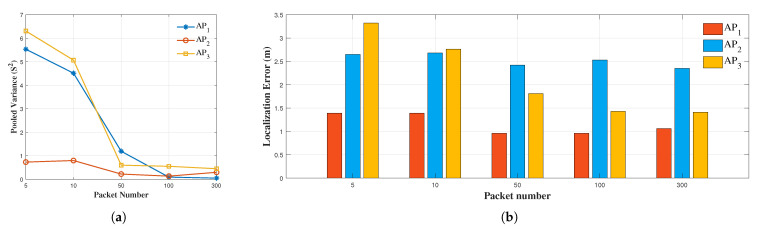
Impact of different input packet numbers on the results. (**a**) Pooled variance of the results with three APs; (**b**) localization error with three APs.

**Figure 10 sensors-22-03709-f010:**
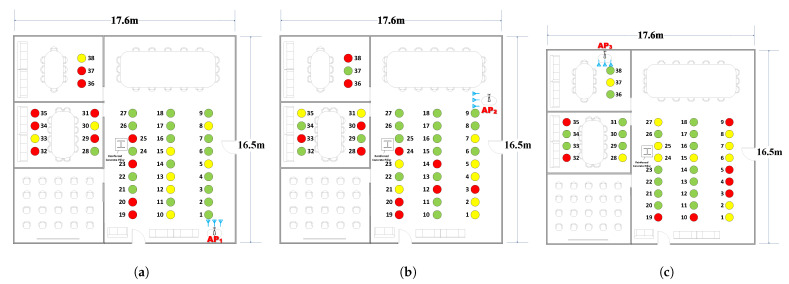
TP-based error map of three different APs. The red circles are BPs, the yellow circles are MA${}_{\mathrm{LP}}$s and the green circles are HA${}_{\mathrm{LP}}$s. The blue symbols show the orientation of the antenna array. (**a**) AP${}_{1}$ has a rich LoS with most locations in the V-shaped sector of the antenna array; (**b**) AP${}_{2}$ has a rich LoS with most locations out of the V-shaped sector; (**c**) AP${}_{3}$ has an NLoS scenario.

**Figure 11 sensors-22-03709-f011:**
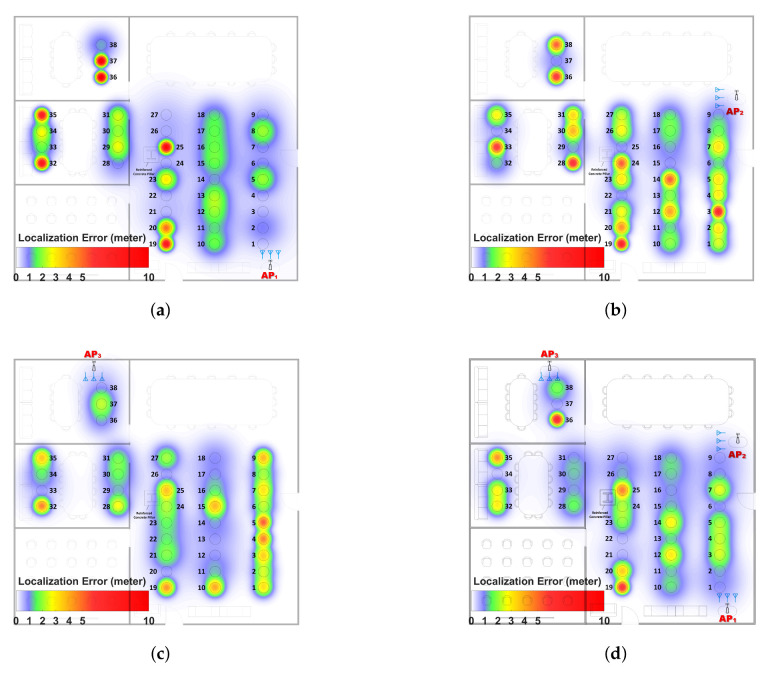
Localization error heat map of (**a**) AP${}_{1}$, (**b**) AP${}_{2}$, and (**c**) AP${}_{3}$ and (**d**) the multi-AP adaptive method. The blue symbols show the orientation of the antenna array.

**Table 1 sensors-22-03709-t001:** Comparison of the RSSI and variance at different locations.

	Location A	Location B
RSSI (dB)	23	19
Variance	0.1	2.1

**Table 2 sensors-22-03709-t002:** Thresholds of HA${}_{\mathrm{LP}}$, MA${}_{\mathrm{LP}}$, and BP.

	AP${}_{1}$	AP${}_{2}$	AP${}_{3}$
TR${}_{1}$	0.9 m	2.3 m	1.4 m
TR${}_{2}$	1.9 m	3.4 m	3.2 m

**Table 3 sensors-22-03709-t003:** HA${}_{\mathrm{LP}}$, MA${}_{\mathrm{LP}}$, and BP in the LoS/NLoS scenario with three APs.

	AP${}_{1}$	AP${}_{1}$	AP${}_{2}$	AP${}_{2}$	AP${}_{3}$	AP${}_{3}$
	(LoS)	(NLoS)	(LoS)	(NLoS)	(LoS)	(NLoS)
BP	4	7	6	5	0	8
MA${}_{\mathrm{LP}}$	6	3	7	2	1	10
HA${}_{\mathrm{LP}}$	17	1	14	4	2	17

## Data Availability

Not applicable.

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
