# Peer review of "Multi-AP and Test Point Accuracy of the Results in WiFi Indoor Localization"

_sensors, 2022, doi:10.3390/s22103709_

Round 1

Reviewer 1 Report

The paper illustrates measurements done with the Intel 5400 NC used for collecting CSI data.  Two algorithms are used for a start: the SpotFi algorithm which is used as benchmark, and the MoLA algorithm, which was previously published by the authors.  Both algorithms are available as free software.  In this paper, as a new contribution, MoLA is adapted to use more than one AP.

The authors use "collaborative" as a synonym for "multi-AP", but this usage is not what is commonly found in the literature, where collaborative systems are those that use data from different users at different times to establish, improve or update a database of reference data, such as fingerprinting, path loss, AP position and so on.  Use of the word "collaborative" in the paper is misleading and should be changed.

The experimental part is lacking rigour and detail.

Most figures uses too many digits of resolution.  Digits of resolution should be consistent with measurement error, and it makes little sense to write figures like 19.49 dB, 10.12 degrees or 1.76 m.  This is an indications of little care for the validity of the presented figures.

Most measurements procedures are not described: how many points were averaged to get the presented measurements, how long was the measurement.  Since the core of this paper is the presentation of results (as indicated in the title), and since results are measures in static position, one would expect great care in defining the credibility of results, for example using confidence intervals, taking measurements in different hours of day and in different days and so on.

Since measurements are independent, it is not clear why the authors use correlation for evaluating the reproducibility.  That should be computed using 1-D or 2-D standard deviation, or 1-D or 2-D quantiles.

Even when using the appropriate statistics, investigating reproducibility (which is one of the main goals of the paper) is a complex business and should involve a well thought-out and accurately described measurement campaign during long time intervals (days, weeks or months).  Nothing like this is done in the paper.

What are the V-shaped sectors?

I suggest significantly reduce the number of pages (for example to six-eight) to only highlight the novel contribution.  The measurement procedure should be improved as noted above.

Reviewer 2 Report

The article extends the Multi-step Optimization Location Algorithm introduced in the article entitled “Multi-Step Optimization of Indoor Localization Accuracy Using Commodity WiFi” by using three access points instead of one and an extra adaptive step in the localization process.

The article has a clear structure and is easy to follow.

I would recommend changing the title of the article or extending the experiment. The reproducibility is only a fragment of the presented research. Furthermore, from my point of view, it is not thoroughly investigated. RSS and AoA in WiFi signals are very susceptible to the surrounding environment. These parameters can vastly change in time and strongly depend on many environmental parameters, which are usually difficult to control. Exhaustive research should involve performing multiple measurements at different times of day/week/traffic/weather etc. If authors consider leaving the title of the article as it is, such an exhaustive analysis should be performed. However, the article also contains some novel ideas for improving the object localization using three access points instead of one, which could be more stressed out in the article, extended, and could be reflected in the title of the article. In such a case, the aspect of reproducibility is sufficiently analyzed.

Authors should explain the background for calculating weights in the multi-access point approach (equations 15-18). It should be proved that it performs better than at least such classical approaches as the Kalman filter, packet filter, unscented filter, and preferably with state-of-the-art ones.

Antenna/Industrial Control Computer (ICC) setup is unclear (line 278-280). Are these antennas placed in different locations in the conference room or the ICCs? How are the antennas mounted? Is ICC housing metal? How are the antennas located versus metal housing? This will undoubtedly influence the measurement results. That should be explained. Maybe some additional photos would explain the details. 

Round 2

Reviewer 1 Report

Most of my observations have been followed and the scope and aim of the paper is now clearer and more consistent with the presented discussion.

Use of a trilateration with Kalman filtering is an improvement.  However, parameters of the trilateration seem arbitrary.  Both the path loss exponent (set to 3) and the received power at 1m (-17 dB) should be justified.

The path loss exponent should be chosen on the basis of the environment at hand, for example by trying different exponents and showing figures illustrating the resulting errors, then selecting the best one for comparison with Mola.  I'd suggest trying exponents from 1.5 to 4 in steps of 0.5.  My impression is that 3 is too high, but I may be wrong and a justification should be given based on measurements.

The received power a 1m should be based on real measured numbers.  If that's the case, it should be stated in the paper, together with the procedure used for measuring it.  While this is an often overlooked issue, this paper is focused on exactly that: measurements, so the issue is important here.

As an alternative, or an addition, a minimisation of the error by jointly varying the exponent and the received power at 1m could be used, and the resulting values for exponent and R1m used in the comparison with mola.

A description of measurements of angle error would enrich the paper.

Reviewer 2 Report

I accept the article in the presented form

Author Response

The reviewer has accepted the article in the presented form.